# Selective nutrient incorporation may underestimate heterotrophy of a mixotrophic reef-building coral
Connor R. Love [1,2] ✉, Marleen Stuhr [3,4], Michael D. Fox [5], Veronica Z. Radice[6], Maoz Fine[4,7], Kelton W. McMahon[1] & David L. Valentine[2,8]

Mixotrophic reef-building corals acquire nutrition via photosynthate translocation from endosymbiotic microalgae and by heterotrophic prey/particle capture. Heterotrophy promotes resistance to, and recovery from, environmental stress, but quantifying coral heterotrophy remains difficult due to complex resource exchanges within the coral holobiont. We interrogated the response of multiple biomarkers to coral heterotrophy using fatty acid profiling and stable isotope analysis of *Stylophora pistillata* grown along a controlled feeding gradient from pure autotrophy to pure heterotrophy. We found that fatty acids and nitrogen were effectively incorporated into both coral host and symbiont tissues, while carbon, which is the primary target for conventional heterotrophy measurements, was not. Our study underscores a functional purpose of heterotrophy to retain essential elements (e.g., N) and molecules (fatty acids) for mixotrophic corals. Selective nutrient incorporation of heterotrophic material also suggests that coral ecologists are likely underestimating the contribution of heterotrophy to a common reef-building coral using conventional carbon isotope offset approaches.

Mixotrophy is an ecologically widespread and flexible strategy in which nutrients can be acquired through autotrophy and heterotrophy. This nutritional approach is found across all major kingdoms and nearly every habitat on Earth, allowing organisms to sustain metabolic demands in changing environments[1]. Mixotrophs act as important nodes for energy flow in food webs[2], with principal nutrient acquisition strategy and plasticity of mixotrophs having large scale impacts on ecosystem structure and global biogeochemical cycles[3]. Reef-building corals are globally important mixotrophs that structure coral reef ecosystems. As such, understanding nutritional sourcing and trophic flexibility of corals is paramount to understanding how reef ecosystems will respond to rapid environmental changes.

The success of tropical reef-building corals has largely been attributed to their symbiosis with the photosynthetic dinoflagellate *Symbiodiniaceae*, in which tight recycling of nutrients between the animal host and photo-endosymbiont (herein 'symbiont') helps the collective coral holobiont meet its metabolic needs[4]. The host and symbiont bidirectionally share energy and organic molecules like amino acids[5], lipids[6], and carbohydrates[7].

Furthermore, inorganic catabolic waste products, such as ammonium, phosphate, and carbon dioxide, are transferred to the symbiont from the host[8] to be fixed back into organic biomolecules by the symbiont. This tight recycling of nutrients within the holobiont leads to high retention of essential nutrients like nitrogen[9], which gives corals a competitive edge in oligotrophic waters[10]. Yet corals have retained the ability to feed heterotrophically through geologic time[11] and often acquire essential biomolecules and elements by feeding on zooplankton and particulate organic matter (POM) in the water column[12,13]. Thus, corals can be involved in multitrophic interactions simultaneously as primary producers, herbivores, carnivores, and detritivores[10,14,15], which makes tracing of material flow and understanding reef food web connectivity difficult.

For several decades, stable isotope ratios of carbon and nitrogen have been used to understand coral trophic strategies in nature[16–18]. In particular, the $\Delta^{13}C$ metric ($\delta^{13}C_{host} - \delta^{13}C_{symbiont}$) has been used extensively as a proxy for coral heterotrophy[16,19]. But this proxy is sensitive to processes beyond heterotrophy that can influence this value. For example, symbiont genotype can affect carbon transfer to the host[20], and variations in assimilation or

[1]Graduate School of Oceanography, University of Rhode Island, Narragansett, RI, USA. [2]Earth Science Department, University of California, Santa Barbara, Santa Barbra, CA, USA. [3]Leibniz Centre for Tropical Marine Research (ZMT), Bremen, Germany. [4]The Interuniversity Institute for Marine Sciences (IUI), Eilat, Israel. [5]Marine Science Program, Division of Biological and Environmental Science and Engineering, King Abdullah University of Science and Technology (KAUST), Thuwal, Saudi Arabia. [6]Department of Biological Sciences, Old Dominion University, Norfolk, VA, USA. [7]Department of Ecology, Evolution and Behavior, The Alexander Silberman Institute of Life Sciences, The Hebrew University of Jerusalem, Jerusalem, Israel. [8]Marine Science Institute, University of California, Santa Barbara, Santa Barbara, CA, USA. ✉e-mail: connor.love@uri.edu

consumption of low $\delta^{13}C$ value lipids can also alter $\Delta^{13}C$ values[21]. Due to the tight recycling of material between the coral host and symbionts, and trophic complexity of reef ecosystems, it is difficult to trace from what source the carbon (or nitrogen) originated[22]. While some studies have used compound-specific amino acid isotope analysis (CSIA-AA) to clarify coral trophic strategies[12,23], the cost and limited availability of instrumentation for such analyses can be prohibitive and lead to low sample throughput. In this study, we aimed to assess the usage of fatty acids (FAs) as an accessible and complementary biomarker to stable isotope approaches for understanding coral mixotrophy.

FAs have been used extensively in qualitative[24] and sometimes quantitative[25,26] assessment of diet in the marine environment. FAs have been used sparingly to qualitatively detect trophic strategy in reef-building coral[13,27–29], though Radice, et al.[28] showed the potential of this tool in quantitatively elucidating >2 diet sources for corals. Due to the sparse use of FAs in coral trophic ecology, despite its potential, there is a critical need for experimentation on the uptake and modification of FAs biomarkers in controlled feeding experiments to better frame results for natural population and ecosystems-scale questions[30].

Here, we conducted an ex situ feeding experiment to evaluate FA and isotopic responses of a common coral (*Stylophora pistillata*) to a gradient in feeding mode: fully autotrophic to fully heterotrophic. This work builds a framework to interpret biomarker responses to changes in coral nutrition through four questions: (1) Does frequency of feeding and bleached or unbleached status alter coral feeding rates? (2) Does holobiont physiological performance reflect shifts in coral feeding mode? (3) Do FA biomarkers in corals reflect changes in source nutrition (autotrophic vs heterotrophic)? (4) How do FA biomarkers patterns compare to bulk tissue carbon and nitrogen isotope patterns for assessing coral feeding? Our study provides new insight into how FA biomarkers and bulk tissue isotopes are recorded into the same coral tissues across a nutritional feeding gradient and what the distinct ecological implications are from these findings.

## Results
### Coral feeding rate changes with food availability
Cumulative *Artemia* nauplii biomass capture (tank-level) increased with experimental feeding regime ($F_{3,6} = 23.71$, $p < 0.001$). Corals fed six times a week (F_6x) ate roughly double the nauplii ($6.61 \pm 1.60$ mg nauplii $cm^{-2}$) that corals fed twice a week (F_2x) ($3.11 \pm 0.83$ mg nauplii $cm^{-2}$) ($p = 0.023$) and showed larger variation in nauplii consumption among tanks (Fig. 1a, Supplementary Table 1). On average, bleached corals fed six times a week (B_F_6x) consumed 19% less nauplii biomass ($p = 0.500$) ($5.36 \pm 1.05$ mg nauplii $cm^{-2}$) over the course of the experiment than unbleached corals fed the same amount (F_6x corals).

Overall, feeding rates of all fed corals varied between ~50–85 nauplii $cm^{-2}$ $h^{-1}$ at a prey density of ~1000 nauplii $L^{-1}$. Feeding rate versus prey density regressions showed that F_2x corals exhibited the highest consumption rates of all experimental treatments ($a = -275.7$, $p < 0.001$; $b = 54.4$, $p < 0.001$; $F_{1, 16} = 40.61$; $R^2 = 0.70$), including over the F_6x condition ($a = -209.1$, $p < 0.001$; $b = 41.0$, $p < 0.001$; $F_{1, 55} = 57.37$; $R^2 = 0.50$) (Fig. 1b). The feeding rate was lowest in bleached corals (B_F_6x, $a = -177.1$, $p < 0.001$; $b = 33.8$, $p < 0.001$; $F_{1, 55} = 41.09$; $R^2 = 0.42$; Fig. 1b).

### Coral physiology responds to feeding
B_F_6x corals generally exhibited significantly lower physiological metrics than all non-bleached coral treatments (Fig. 2, Supplementary Table 2). Feeding by non-bleached corals (F_2x and F_6x) generally increased mean physiological metrics compared to unfed corals (control), although this effect was only statistically significant for chlorophyll, skeletal (aragonite) growth, and maximum electron transport rate (Fig. 2e, f, h and Supplementary Table 2).

Total host FA mass was 75% lower in B_F_6x corals than control corals (p < 0.001, Supplementary Table 2) but remained stable in F_2x and F_6x treatments (Fig. 2a, Supplementary Table 2). Similarly, in the symbiont fraction, total symbiont FA mass did not significantly change with increased

feeding in non-bleached corals (Fig. 2b, Supplementary Table 2). Host protein increased slightly with feeding (Fig. 2c, Supplementary Table 2) but decreased in B_F_6x corals by 26% relative to control corals (Fig. 2c, Supplementary Table 2). Symbiont protein remained stable across treatments (Fig. 2d). B_F_6x corals exhibited significantly lower, near zero, aragonite growth (0.1 mg $cm^2$ $d^{-1}$, $p = 0.03$), while fed, non-bleached corals (F_2x and F_6x) nearly doubled calcification rates relative to control corals (F_2x: $p = 0.02$, F_6x: $p = 0.03$; Fig. 2e, Supplementary Table 2). B_F_6x corals exhibited significantly lower photo-physiology metrics ($rETR_{max}$ and $F_v/F_m$) compared to control corals ($p < 0.001$, Fig. 2f, Supplementary Fig. 1). The effect of feeding on non-bleached corals was not statistically significant for $F_v/F_m$ (Fig. S1, Table S3), though maximal electron transport rate did increase significantly by ~25% relative to control corals (F_2x: $p = 0.021$; F_6x: $p = 0.029$, Supplementary Fig. 1, Supplementary Table 3). Time was a significant predictor in linear mixed effects models of $F_v/F_m$ and $rETR_{max}$ (Supplementary Table 3), which was driven primarily by B_F_6x corals (Supplementary Fig. 1). In B_F_6x corals, average photo-physiology metrics declined over the course of the experiment despite continuous feeding for the duration of the experiment and high final host protein.

B_F_6x corals exhibited lower symbiont densities than control corals ($p = 0.067$) while F_2x and F_6x exhibited slightly higher mean symbiont densities than control corals, though the effect was non-significant (Supplementary Table 2). A similar trend was seen for holobiont total chlorophyll, in which B_F_6x corals exhibited ~50% less chlorophyll than control corals ($p = 0.039$), while F_2x (58% increase, $p = 0.0244$) and F_6x (95% increase, $p = 0.002$) showed significant increases in total chlorophyll compared to control corals (Supplementary Table 2).

### Heterotrophy alters fatty acids in host and symbiont
Shifts in coral heterotrophy were more accurately recorded in relative abundance data of FAs (% of total) than mass normalized data ($\mu g$ $g^{-1}$ dry tissue) in both the host and symbiont fractions. Linear mixed effects models of relative abundance host fatty acids revealed 43 significant fixed effect coefficient estimates ($p < 0.05$, Supplementary Table 4) as compared to 33 for mass normalized data (both data types showed changes in a total of 25 FAs). Similarly, in the symbiont fraction, there were 24 significant fixed effect factor estimates ($p < 0.05$) across 17 FAs for relative abundance data (Supplementary Table 5), while mass normalized data exhibited only 17 significant fixed effect estimates across 14 FAs. Thus, all fatty acid data herein are presented as relative abundance data.

Experimental nutritional source groups (heterotrophic source = *Artemia* nauplii, autotrophic source = control coral symbionts) completely separated via principal component analysis (Fig. 3a), with 24 of 27 FAs showing significant differences ($p < 0.05$) between source groups (19 of which had $p < 0.01$; Wilcoxon rank sum test, see Supplementary Table 6). FAs that had significantly higher relative abundance in nauplii ($p < 0.05$) are hereafter referred to as "heterotrophic biomarkers" and FAs that were significantly higher in relative abundance in the autotrophic source (control coral symbionts) are hereafter referred to as "autotrophic biomarkers".

There were four emergent patterns in coral host FA profiles due to changes in heterotrophy: positive correlation with heterotrophy, negative correlation with heterotrophy, bleaching effect, and no effect (Fig. 3b–f). Most biomarker patterns generally tracked the source group. For example, 18:2n6, 18:3n3, and 20:5n3 were all heterotrophic FA biomarkers that significantly increased with heterotrophy relative to control corals and 18:3n6, 20:1n9, 20:3n6, 22:1n9, and 23:0 were all autotrophic FA biomarkers that significantly decreased with increasing heterotrophy (Fig. 3, Supplementary Table 4). However, not all FA biomarkers followed this simple trend. For example, the autotrophic FA biomarker 22:4n6 significantly increased with increasing heterotrophy (Supplementary Table 4), and the heterotrophic FA biomarker 18:1n9 significantly decreased with increased heterotrophy (Supplementary Table 4 and Supplementary Fig. 2). FAs like 16:1n7 showed a bleaching effect where there was a significant change between fed, bleached (B_F_6x) and fed, unbleached (F_6x) corals, with little to no change due to feeding level (Fig. 3b, e). Most FAs that exhibited a bleaching effect pattern

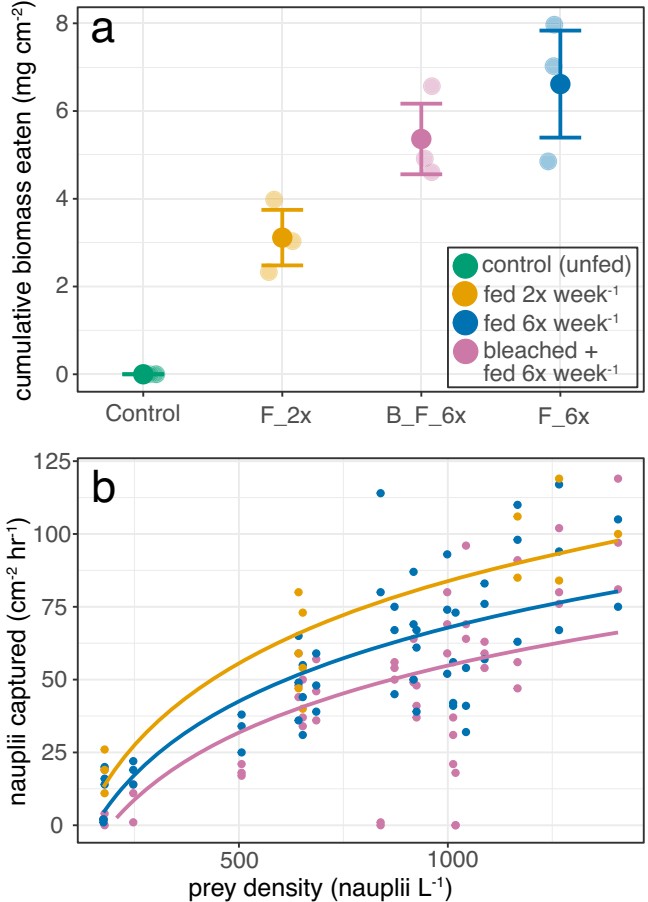

**Fig. 1 | Coral feeding rates scale with food availability but are reduced in bleached corals.** *Stylophora pistillata* feeding rates and prey density showing (**a**) cumulative biomass capture for the duration of the experiment (error bars represent standard errors for $n = 3$ independent tanks) and (**b**) surface area normalized nauplii capture rates relative to prey density, with a logarithmic regression fit for each treatment. Each experimental group (control: green, fed 2x week$^{-1}$: yellow, fed 6x week$^{-1}$: blue, bleached + fed 6x week$^{-1}$: pink) represents three tanks, and each tank cumulative biomass capture represents an average surface area normalized parameter for the ~10 coral fragments in each tank. Control corals are represented in green.

were autotrophic markers declining with the B_F_6x treatment, but three saturated FA, including 18:0, 22:0 (heterotrophic markers), and 20:0 (a non-distinguishing biomarker), all significantly increased with the B_F_6x treatment (Fig. 3b, Supplementary Table 4). The remaining 9 FAs showed no discernable trends across experimental treatments (Fig. 3b, f).

Like coral host tissues, coral symbiont tissues responded to shifts in coral heterotrophy as well. The heterotrophic FA biomarkers 18:3n3, 18:1n7 and 22:0 increased significantly in the symbiont fraction with increasing heterotrophy (Supplementary Table 5). Interestingly, two heterotrophic FA biomarkers, 18:2n6 and 20:4n6, decreased in the symbionts with increasing heterotrophy (Supplementary Table 5), although the effect was only significant ($p < 0.05$) for 18:2n6 in the F_6x treatment ($p = 0.018$). Several autotrophic FA biomarkers (14:1, 18:4n3 and 22:6n3) increased significantly in the symbiont fraction with feeding ($p < 0.05$, Supplementary Table 5), while several other autotrophic markers (14:0, 16:1n7, and 20:3n6), decreased significantly with increasing heterotrophy (Supplementary Table 5 and Supplementary Fig. 2).

### Fatty acid and isotope biomarkers reveal low heterotrophic carbon incorporation
Nutritional source group $\delta^{15}$N values were $-1.5 \pm 0.4$‰ for the autotrophic source (symbionts of control corals) and $9.8 \pm 1.0$‰ for the heterotrophic

source (*Artemia* nauplii). The $\delta^{15}$N values of host and symbiont fractions increased significantly towards the heterotrophic source value as heterotrophy increased (Host: F_2x: $+2.1$‰, $p < 0.001$, F_6x: $+3.7$‰, $p < 0.001$, B_F_6x: $+3.7$‰, $p < 0.001$; Symbiont: F_2x: $+2.2$‰, $p < 0.001$, F_6x: $+3.5$‰, $p < 0.001$; Fig. 4a, Supplementary Tables 4 and 5). The molar carbon to nitrogen ratio (C:N) was $4.48 \pm 0.14$ for the heterotrophic source and $5.79 \pm 0.36$ for the autotrophic source. The C:N ratio also significantly changed towards the heterotrophic source value with feeding (Host: F_2x: $-0.17$, $p = 0.012$, F_6x: $-0.30$, $p < 0.001$, B_F_6x: $-0.68$, $p < 0.001$; Symbiont: F_2x: $-0.25$, $p = 0.035$, F_6x: $-0.61$, $p = 0.001$; Fig. 4c), with the C:N ratio for B_F_6x corals ($4.41 \pm 0.15$) converging with the C:N value of the heterotrophic source value ($4.48 \pm 0.13$). There was no significant effect of feeding on the difference between host and symbiont nitrogen isotope ratios ($\Delta^{15}$N, F = 1.1346, $p = 0.3878$) although the average host fraction $\delta^{15}$N value was consistently ~2‰ larger than the symbiont fraction (Fig. 4a, Supplementary Tables 4 and 5).

The $\delta^{13}$C values of host and symbiont fractions generally decreased with heterotrophy towards the heterotrophic source signal ($\delta^{13}C_{nauplii} = -20.5 \pm 1$‰), but the effect was non-significant despite a ~4.4‰ difference between nutritional sources (Host: F_2x: $-0.46$‰, $p = 0.147$, F_6x: $-0.39$‰, $p = 0.203$, B_F_6x: $-0.391$‰, $p = 0.211$; Symbiont: F_2x: $0.00$‰, $p = 0.967$, F_6x: $-0.13$‰, $p = 0.542$, Fig. 4b, Supplementary Table 6). The difference between host and symbiont carbon isotope ratios ($\Delta^{13}$C) decreased in mean value per treatment with increasing heterotrophy but the effect was non-significant ($p = 0.5117$, F = 0.6755; Fig. 4e). Additionally, $\Delta^{13}$C value did not strongly correlate ($R^2 < 0.16$) with any other measured parameters other than $\delta^{13}C_{host}$ ($R^2 = 0.52$), showing that a few fragments with very low $\Delta^{13}$C values ($< -2$‰) were driving most of this trend (Fig. 4f).

B_F_6x corals hosts turned over $33.6 \pm 7.5$% of their total nitrogen with heterotrophically acquired nitrogen (Supplementary Table 7). F_6x and F_2x coral hosts turned over $38.1 \pm 7.3$% and $22.4$% of their heterotrophically acquired nitrogen, respectively, while F_6x symbionts and F_2x coral symbionts turned over ~31% and ~19% of their heterotrophically acquired nitrogen, respectively. For carbon, ~9.5% of host tissue carbon was replaced with heterotrophically acquired carbon for B_F_6x and F_6x corals. F_6x coral symbionts turned over less carbon than the host with only ~5.4% of its carbon biomass replaced with heterotrophic carbon. For the F_2x treatment, there was a similar ~ 9.5% of host carbon replaced with heterotrophic carbon and a ~1.5% replacement of symbiont carbon. Considering the C:N ratio of ~4.5 for the heterotrophic source (9 atoms of carbon for every 2 atoms of nitrogen, Supplementary Table 6), this results in a preferential integration of nitrogen into biomass by a factor of ~16 for F_6x and B_F_6x host (i.e., ~16 atoms of heterotrophic nitrogen are measured in host biomass for every 1 atom of heterotrophic carbon), a factor of ~26 for F_6x symbionts, a factor of ~9 for F_2x host, and a factor of ~60 for F_2x symbionts.

## Discussion
We show that FA biomarkers coupled with nitrogen isotopes are a promising tool to better understand coral mixotrophy. Several FAs ($n = 10$) showed clear and significant patterns of incorporation with heterotrophy in a controlled experiment and heterotrophic nitrogen exhibited preferential incorporation by a factor of ~10–60 over heterotrophic carbon, suggesting that carbon isotopes were not a reliable indicator of heterotrophy. These findings suggest that coral heterotrophy in some species may serve mainly to supplement the holobiont with essential biomolecules (i.e., fats and amino acids) and elements like nitrogen that can be retained via tight nutritional recycling with the coral symbiont *Symbiodiniaceae*, while carbon is likely respired or exuded as mucus[7]. Coral heterotrophy is shown to respond to natural variations in food supply[31], making this strategy a strong advantage in oligotrophic waters and environments with changing resource availability[32,33] where it can help corals recover from[19,34,35], and even resist, environmental stresses like bleaching[36]. Our study provides valuable tools to identify this heterotrophic feeding strategy in corals.

https://doi.org/10.1038/s42003-025-08621-8                                                                    **Article**

**Fig. 2 | Physiological metrics positively scale with heterotrophy but do not offset bleaching effects.**
Physiology metrics by experimental treatment (bleached + fed 6x week$^{-1}$: pink, control: green, fed 2x week$^{-1}$: yellow, fed 6x week$^{-1}$: blue) are plotted as points with each point representing a coral fragment with box and whisker plots added, horizontal lines in box represent median and quartiles while whiskers represent 1.5(IQR) distance from each upper and lower quartile, horizontal connecting bars between treatments indicate significance levels of post hoc pairwise contrasts: *$p < 0.05$, **$p < 0.01$, ***$p < 0.001$. **a** FA mass in host tissues normalized to surface area (B_F_6x: $n = 29$, control: $n = 23$, F_2x: $n = 22$, F_6x: $n = 27$), **b** FA mass in symbiont normalized to dry tissue mass (control: $n = 23$, F_2x: $n = 21$, F_6x: $n = 27$), (**c**) surface area normalized host protein (B_F_6x: $n = 33$, control: $n = 31$, F_2x: $n = 27$, F_6x: $n = 33$), **d** symbiont protein cell$^{-1}$ (control: $n = 28$, F_2x: n = 24, F_6x: $n = 31$), **e** whole experiment aragonite skeleton growth (B_F_6x: $n = 31$, control: $n = 27$, F_2x: $n = 25$, F_6x: $n = 32$), **f** relative electron transport rate from day 15 of the experiment as a representative time point of patterns seen in photo-physiology (B_F_6x: $n = 32$, control: $n = 29$, F_2x: $n = 27$, F_6x: $n = 33$, see supplementary information for more details), **g** symbiont density normalized to surface area (B_F_6x: $n = 33$, control: $n = 28$, F_2x: $n = 24$, F_6x: $n = 30$), and (**h**) total chlorophyll concentration normalized to surface area (B_F_6x: $n = 33$, control: $n = 31$, F_2x: $n = 27$, F_6x: $n = 33$).

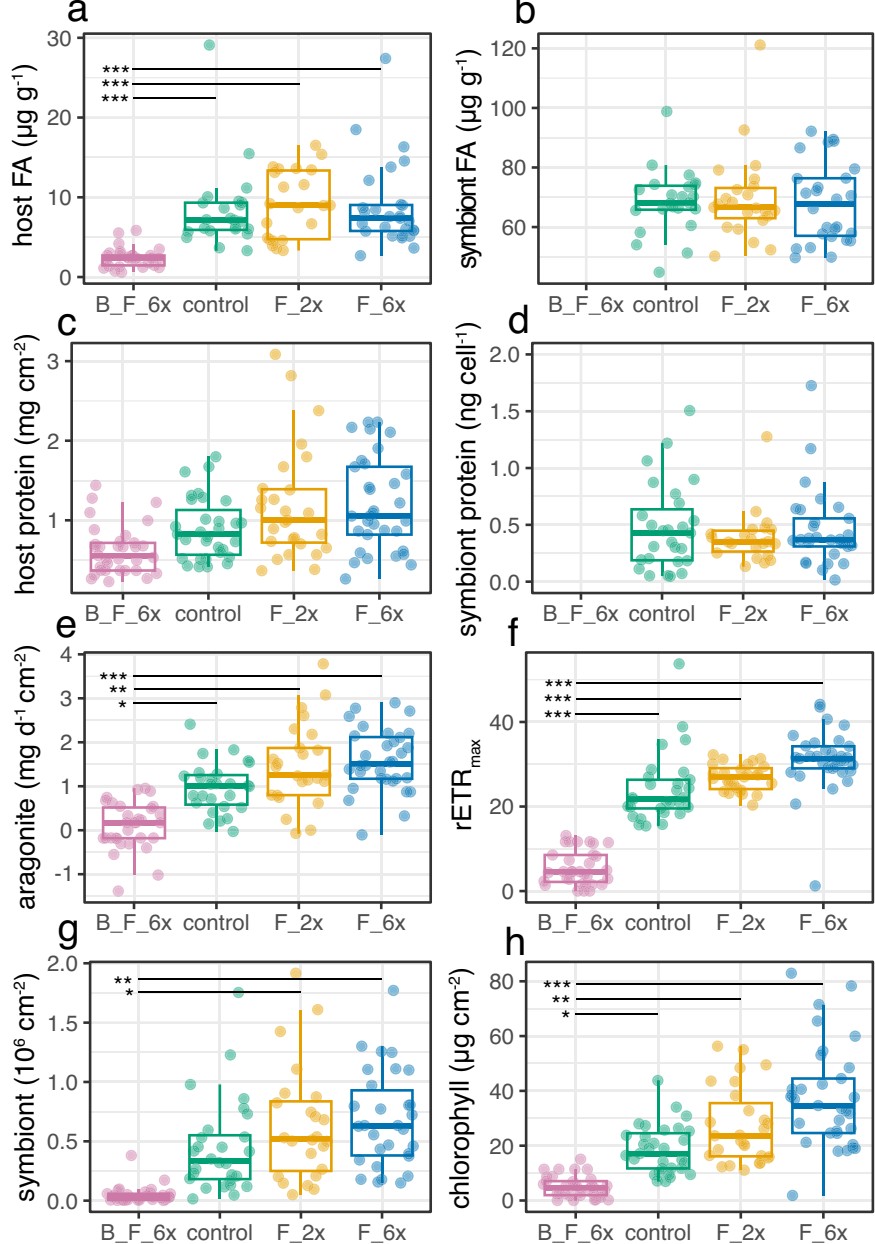

A gradient in cumulative nauplii feeding was observed during the experiment (Fig. 1a). All corals ate at a rate between ~50–85 nauplii cm$^{-2}$ h$^{-1}$ at ~1000 nauplii L$^{-1}$, which is similar to but slightly lower than a previous study of Red Sea *S. pistillata* (103 nauplii cm$^{-2}$ h$^{-1}$; Hoogenboom et al., 2015). This may be, at least partially, explained by differences in experimental feeding conditions (flow chamber versus tank). Logarithmic regressions per treatment showed that F_2x corals exhibited higher consumption rates than F_6x condition, suggesting that *S. pistillata* may reach a feeding saturation in which feeding rate declines as number of *Artemia* nauplii captured per day increases. This stands in contrast to results for Gulf of Aqaba *S. pistillata* in which corals fed 2x week$^{-1}$ and 6x week$^{-1}$ consumed natural zooplankton at similar rates[37]. The apparent feeding saturation we observed is likely not reached in nature where planktonic densities are much lower (~ 1–2 orders of magnitude lower in biomass m$^{-3}$ in oceanic water, ~ 2–3 orders of magnitude lower in lagoonal reef waters[38]). Bleached corals (B_F_6x) exhibited the lowest feeding rates of any treatment (Fig. 1b), even though cumulative biomass consumption was not statistically different (Fig. 1a, see Results). That said, mean cumulative biomass consumption of

bleached corals was still lower than non-bleached corals fed the same amount (Fig. 1a). Altogether, this suggests (with soft evidence) that some corals can exhibit reduced feeding rates after bleaching[39] and that there may be an energetic cost to feeding that is supplemented by the symbionts. One potential explanation is that since Red Sea *S. pistillata* can generate ATP from stored carbohydrates[40] that are mainly generated by the symbiont[7], this would reduce the energetic pool available to bleached corals for feeding activities (e.g., tentacle movements). Although our results are contrasting, reduced feeding rates of bleached corals would have further implications in the face of a warming global ocean. Here, bleached corals would consume less plankton than unbleached corals given the same heterotrophic food supplies, further amplifying the negative effects of coral bleaching on some coral species. This is a hypothesis ripe for further investigation.

Generally, physiological parameters positively scaled with heterotrophy. For example, feeding significantly increased relative electron transport rate (rETR$_{max}$) and chlorophyll concentration (Fig. 2f, h), showing that feeding incurred a positive feedback loop on some symbiont related physiology metrics. Skeletal growth also positively scaled with heterotrophy;

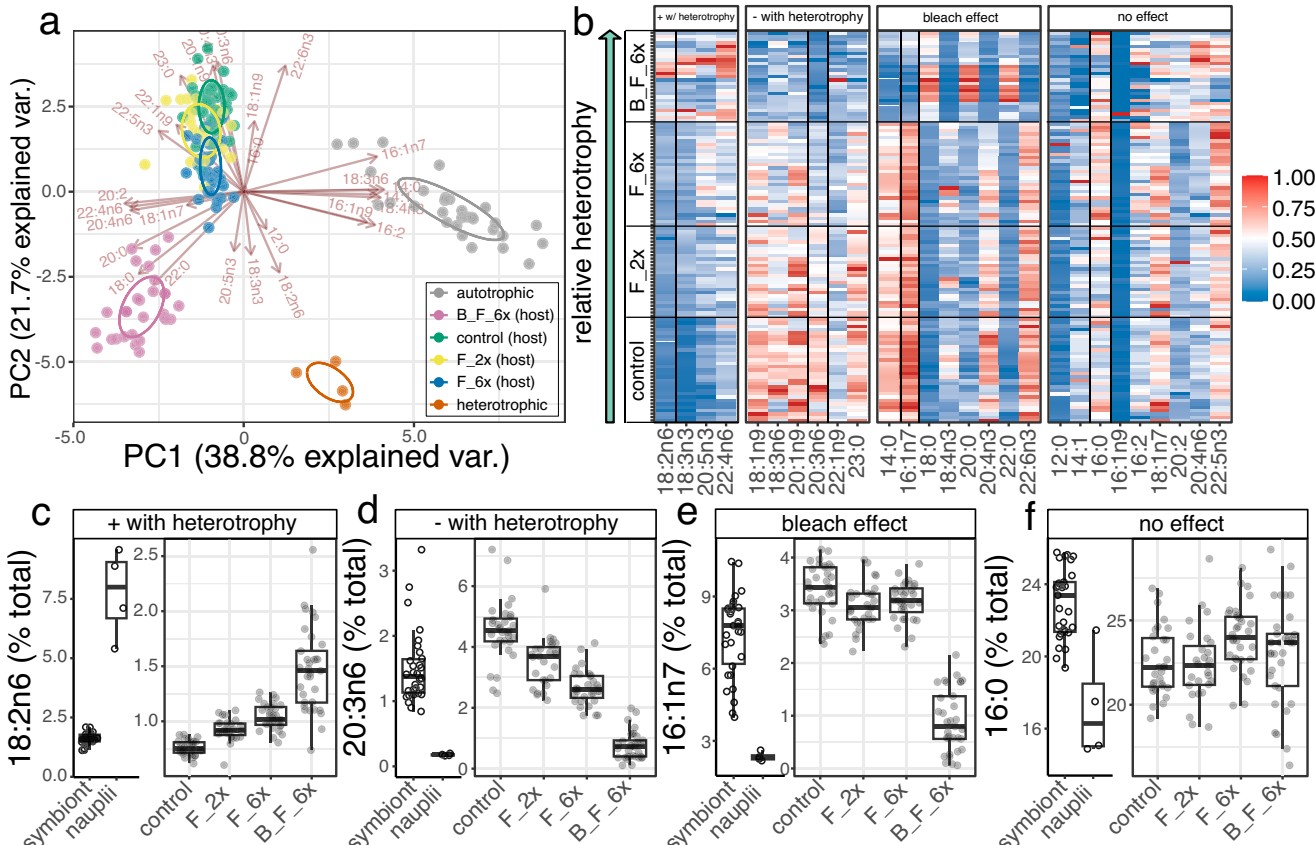

**Fig. 3 | Nutritional source groups completely separate via their fatty acid profiles and cause four distinct incorporation patterns in coral host tissues across a nutritional gradient.** Differences in fatty acids of nutritional source groups recorded in coral host tissues with four principal patterns, visualized by (**a**) a principal component analysis of nutritional source groups with all fatty acids, using a reduced number of displayed FA vectors for clarity, and (**b**) a heatmap of min-max normalized FA data (blue to red scale bar from 0 [minimum, blue] to 1 [maximum, red]) for all corals, grouped by experimental source group on the y-axis and observed pattern of individual FAs on the x-axis. Representative FAs for each observed pattern are outlined in bold and plotted in **c**–**f**. For panels **c**–**f**, each data point represents a unique coral fragment (control: $n = 31$, F_2x: $n = 27$, F_6x: $n = 32$, B_F_6x: $n = 33$ and $n = 31$ for symbiont source data) or represents a unique batch of hatched nauplii ($n = 4$ for nauplii source data). Data points in **c**–**f** show the percent total for symbiont and nauplii sources, as well as coral host tissues across treatment groups and are plotted with associated box and whisker plots. Horizontal lines in box represent median and quartiles while whiskers represent 1.5(IQR) distance from each upper and lower quartile. **c** 18:2n6 as a representative FA that increased with heterotrophy in coral host tissues, (**d**) 20:3n6 as a representative FA that decreased with increasing heterotrophy, (**e**) 16:1n7 as a representative FA that decreased only in bleached corals, and (**e**) 16:0 as a representative FA that did not significantly change during the course of the experiment.

and was net even positive for unfed control corals (Fig. 2e), perhaps due to the resilient nature of Red Sea *S. pistillata*[41], short experiment duration (3 weeks), and/or some amount of heterotrophy on ≤130-μm particles let in by the sweater supply system (see Methods). Overall, feeding did not offset negative bleaching effects (Fig. 2). Although it is known that the menthol induced bleaching approach we used does result in different mechanisms of symbiont removal than thermal bleaching in anenomes[42], the physiological and biochemical performance of *S. pistillata* does not significantly change with menthol bleaching compared to unbleached corals[43]. As such, we interpret patterns of our bleached and fed corals (B_F_6x) to stem primarily from the observed low symbiont densities (Fig. 2g, Supplementary Table 2), low photo-physiological performance (Fig. 2f, Supplementary Table 3), and cumulative nauplii biomass consumed (Fig. 1a, Supplementary Table 1) rather than bleaching method. While total FA mass in unbleached coral host tissue (control, F_2x, and F_6x) remained relatively stable, it decreased by 75% in B_F_6x corals. This pattern suggests that the symbiont contributes large quantities of FAs to the host that cannot be replaced by heterotrophic feeding alone[44–46]. In the non-bleached feeding treatments, there was remarkable stability of total FA mass despite large differences in feeding regime. This FA stability has been observed in total lipid studies before[29,47], but there is also contrasting evidence that fats increase with feeding[48]. The increase in protein in both host and symbiont of non-bleached corals with

feeding, although non-significant ($p > 0.05$), has been observed as well[49,50], while the significant decrease in protein in bleached and fed corals (by 26%) has been observed in natural bleaching events[51]. Interestingly, we found that total FA mass of B_F_6x host tissues decreased more than total protein (75% vs 26%, respectively). This likely indicates that bleached corals were preferentially catabolizing FAs for energy to compensate for low symbiont densities[17,52,53], but may also include some routing of lipid reserves into amino acid synthesis[50].

Changes in coral heterotrophy elicited significant changes in many FAs in both the host (Fig. 3b–f, Supplementary Table 4) and symbiont fraction (Supplementary Table 5) due to a strong difference in source FA profiles (Fig. 2a, Supplementary Table 6). Four distinct patterns were observed in host tissues (Fig. 2). The heterotrophic FA biomarkers 18:2n6 and 18:3n3, the building blocks of n6 and n3 PUFAs that presumably cannot be made by the host de novo[54,55], significantly increased with heterotrophy. 18:3n3 is often close to or undetectable in starved corals[28,56] (Supplementary Table 4), and our results add to growing evidence that this FA may be sourced almost entirely from heterotrophy in *S. pistillata*[44,48]. However, it has also been shown that some species of *Acropora* exhibit remarkably high 18:3n3 values as the most predominant PUFA, suggesting chemotaxonomic and/or ecological differences between *Stylophora* and *Acropora*[46,57]. The low relative abundance of 18:3n3 in *S. pistillata*, even in high feeding treatment corals (<

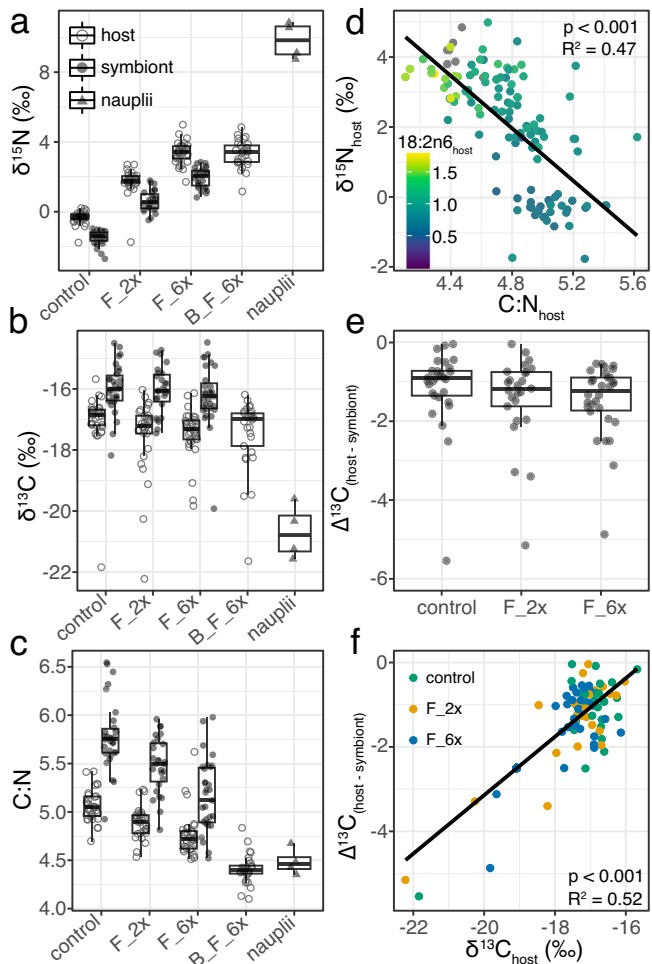

**Fig. 4 | Nitrogen isotopes, carbon and nitrogen ratio, and select fatty acids scale with heterotrophy while carbon isotopes do not.** Panels **a**–**c** show coral host and symbiont tissue data separated by experimental treatment (Host fraction- control: $n = 31$, F_2x: $n = 29$, F_6x: $n = 32$, B_F_6x: $n = 29$; Symbiont fraction- control: $n = 30$, F_2x: $n = 27$, F_6x: $n = 32$) as well as heterotrophic source group *Artemia* nauplii ($n = 4$ unique batches of hatched nauplii). Panels **a**–**c** and panel **e** are plotted with box and whisker plots, horizontal lines in box represent median and quartiles while whiskers represent 1.5(IQR) distance from each upper and lower quartile. **a** $\delta^{15}$N values. **b** $\delta^{13}$C values. **c** C:N ratios. **d** $\delta^{15}$N value vs. C:N ratio of host tissue, colored by relative abundance of essential FA 18:2n6. **e** $\Delta^{13}$C values ($\delta^{13}$C$_{host}$ - $\delta^{13}$C$_{symbiont}$) grouped by experimental treatment with bleached condition removed (control: $n = 30$, F_2x: $n = 25$; F_6x: $n = 32$). **f** $\Delta^{13}$C values plotted against $\delta^{13}$C$_{host}$ values, colored by experimental treatment.

2% of total FAs), suggests this is a vital metabolic precursor molecule that is rapidly modified into other FAs, such as 22:6n3 or 20:5n3, once it is consumed.

There was a divergent pattern observed within autotrophic biomarkers, where some decreased with heterotrophy (e.g., 18:3n6), as expected, while others surprisingly increased with heterotrophy (e.g., 22:5n3) (Fig. 3, Supplementary Table 4). We interpret this divergent pattern to be a result of "slow" and "fast" turnover pools of symbiont-derived FAs. Fast turnover FAs, like 22:5n3, scale with symbiont density, chlorophyll concentration, and symbiont photosynthesis that are known to increase with enhanced feeding[58], while slow turnover FAs, like 18:3n6, are "diluted" by the contribution of heterotrophically-sourced FAs. At this time, we do not know if this pattern is consistent across different coral species and symbiont types but nonetheless represents an important finding to better help interpret FA profiles found in nature.

Bleaching resulted in significant decreases in several PUFAs (e.g., 22:6n3, 18:4n3 and 20:4n3) and a consequent increase in several saturated FAs (e.g., 18:0, 20:0 and 22:0, Fig. 3b, e), which has been observed in *Acropora* corals as well[59,60]. 14:0,16:1n7, 18:4n3, 20:4n3 and 22:6n3 showed dramatic declines in the B_F_6x treatment (Fig. 3b, Supplementary Table 4), which is in line with evidence that the PUFAs in this list are mainly sourced from the symbionts[61]. The sourcing of 14:0 and 16:1n7 from the symbionts is less defined. Nonetheless, these FAs are often found in higher proportions in the symbionts than the host of several species[46] (Supplementary Tables 4 and 5) and follow patterns of symbiont sourcing (Fig. 3b). Overall, host FA profiles systematically shifted towards the heterotrophic source end member (Fig. 3a), with the host of bleached corals showing no overlap in FA profile with non-bleached corals (Fig. 3a). This suggests that the absence of symbionts and/or the negative impacts of bleaching were larger than the positive effect of feeding, resulting in distinguishable FA profiles between bleached and non-bleached corals (Fig. 3a).

Nitrogen isotope values significantly increased towards the heterotrophic source in host and symbiont tissues nearly equally (Fig. 4a, c, Supplementary Table 4) and correlated strongly with essential FAs like 18:2n6 (Fig. 4d and Supplementary Figs. 3 and 4). Conversely, carbon isotope ratios and $\Delta^{13}$C values did not significantly change with feeding in ~90% of the fragments (Fig. 4b, e, f, Supplementary Fig 3, Supplementary Table 4). As such, carbon isotopes and the $\Delta^{13}$C proxy "captured heterotrophy" ($\Delta^{13}$C < $-2$‰) in only ~13% of the fragments (Fig. 4f). When calculated as percent incorporation with heterotrophically derived nitrogen, heterotrophic nitrogen exhibited preferential incorporation by a factor of ~10–60 over heterotrophic carbon. This suggests that these corals were not carbon limited and instead selectively incorporated nitrogen[62,63] and other select FAs into their tissues. We interpret this to mean that heterotrophy by *S. pistillata* is suited mainly to supplement the holobiont with key limiting elements (e.g., nitrogen) and essential FAs that cannot be made de novo in significant quantities[54,64], whereas carbon may have been respired or exuded as mucous[7]. This finding, coupled with no strong signal in $\Delta^{13}$C suggests that coral heterotrophy in nature may be historically underestimated based on conventional interpretation of bulk isotope data if heterotrophic carbon is not effectively recorded into coral tissues and source group nitrogen isotope ratio differences are small[65,66] (Supplementary Fig 5). Our study underscores the value of using complementary tracer approaches in elucidating nutritional sourcing in mixotrophic organisms and finds that selective nutrient incorporation may be causing a considerable underestimate of the contribution of heterotrophy to corals in nature.

## Methods
### Coral collection, experimental setup, and feeding rate measurements

Fragments of *S. pistillata* (5–8 cm) were collected from 12 colonies ($n = 7$–16 fragments colony$^{-1}$) in the Interuniversity Institute for Marine Sciences underwater nursery on 11/20/19 and 11/28/19 at ~5 m depth. Corals were acclimated to tank conditions (ambient seawater temperature, $200 \pm 91$ μmol quanta m$^{-2}$ s$^{-1}$) in the Red Sea Simulator[67] for 10 d and then randomly assigned to three replicate 40 L tanks ($n = 8$–11 fragments tank$^{-1}$) per treatment for 22 d: (1) unfed "Control", (2) fed 2x per week "F_2x", (3) fed 6x per week "F_6x", (4) bleached and fed 6x per week "B_F_6x". Two additional tanks were kept with no corals to sample background system POM. After 3 d of tank acclimation, the 'bleaching condition' corals were bleached using shaken menthol-DCMU incubations for 4 days[43,68]. After the 4 d incubations, coral fragments were visibly white and appeared to have polyps fully extended within 3 d after bleaching. Tanks were equipped with a small pump (Aqua One Maxi 101, 400 L h$^{-1}$) to maximize water circulation and received flow-through 130 μm filtered natural seawater.

Fed corals received *Artemia* nauplii that were hatched daily from a singular egg source (Eilat Underwater Observatory) daily. We aimed to feed corals at a nauplii density of 1000 nauplii L$^{-1}$. Variations in hatching temperatures resulted in variations in total nauplii stock available, and thus

nauplii stock was counted in triplicate daily just prior to feeding. Corals were fed for 2 h, during which time tank flow was halted, though water pumps remained on to evenly distribute nauplii. Triplicate water samples were taken immediately after feeding to assess post feeding nauplii density, and then tank flow was restored to flush remaining nauplii. The density of remaining nauplii in each tank was measured in triplicate. Counting was performed with a microscope and a plexiglass plankton counting tray (General Oceanics, product 1810-B10). The difference between initial and final nauplii density was used to calculate nauplii consumed per tank for each feeding, numerical nauplii consumption data was converted to biomass with an average nauplii mass of 3.2 μg[69]. Due to variations in nauplii stock availability, and thus prey concentration, we were able to create prey density versus consumption rate plots and models (Fig. 1b).

Corals were flash frozen in liquid nitrogen ~24 h after the last feeding and airbrushed with 10 mL of cold phosphate buffer (0.1 M) with EDTA (0.1 mM, pH = 7.0) at 4 °C and manually homogenized on ice. Separation of the host and symbiont fractions was achieved through centrifugation (3000 g for 5 min at 4 °C), and the host fraction (supernatant) was decanted while the endosymbiont pellet was resuspended in 25% of the original volume of phosphate buffer and centrifuged again. The supernatant of this second centrifugation step was added to the host fraction and this was centrifuged again to remove any remaining endosymbiont cells. Both fractions were lyophilized at −80 °C and stored with $N_2$ gas headspace in each vial for preservation of polyunsaturated fatty acids (PUFA).

### Particulate organic matter and zooplankton sampling

In situ POM samples (10 L filtered onto pre-combusted 0.7 μm GF/F filters) were collected ~ weekly at 5 m depth near the coral nursery via 5 L Niskin bottle weekly, starting 1 week before the experiment, to characterize a particulate heterotrophic source prior to the experiment. POM samples from two tanks without corals were also collected on the same day to constrain the chemical and isotopic values of ≤130-μm filtered POM entering the tanks.

Natural zooplankton populations near the coral nursery were sampled for biochemical comparisons to the experimental nauplii. Near-reef plankton (200 μm mesh net) were collected from the pier adjacent to the coral nursery overnight to best mimic the known coral feeding times and highest densities of plankton in the water column above the reef. Pelagic plankton were sampled along the 350 m isobath (100 μm mesh net) at 20 m depth at ~14:00, local time. *Trichodesmium* colonies and planktonic foraminifera were picked out of the subsequent samples to avoid contaminating the end member biochemical values with non-standard diet sources. All plankton samples were immediately frozen at −80 °C for lyophilization.

### Fatty acid extraction and analysis

Coral host (15 mg) and symbiont (3–5 mg) samples were extracted using a modified Folch method[70] following Taipale et al.[71] and Radice et al.[28]. 2-methyldodecanoic acid (C12- methyl branched) and nonadecenoic acid (C19:1) were used as internal standards for mass normalization. FAs were analyzed with a Gas Chromatograph (GC) equipped with a Flame Ionization Detector (GC-FID, Hewlett Packard HP5890) and a Supelco Omegawax 250 Column (30 m, 0.25 mm ID, 0.25 μm film thickness) (see Supplementary Table 8 for temperature ramp parameters and Supplementary Table 9 for gas flow parameters). FAs were identified by: (1) comparison of retention times and peak area to a certified reference material (Supelco 37 component FAME mix, FAME-37), (2) spiking experiments with known analytes, and (3) analyzing a representative subset of samples on a GC equipped with a mass spectrometer (GC-MS). Mass of fatty acid per sample was calculated by dividing peak area by a daily calibrated response factor for that compound from a standard mix (Supelco FAME-37). Analytical precision for relative abundance data (calculated from FAME-37) was ± 0.04% and precision for mass normalized data was ~0.1 μg g⁻¹.

### Isotope ratio measurements

Freeze-dried tissues were acidified with 6% sulfurous acid to remove any inorganic carbonates. Samples were analyzed for $\delta^{13}C$ and $\delta^{15}N$ values using a Thermo Finnigan Delta-Plus Advantage isotope ratio mass spectrometer coupled with a Costech EAS elemental analyzer in the University of California Santa Barbara Marine Science Institute Analytical Laboratory. Instrument calibration was conducted using acetanilide reference standards. Instrument precision was determined using replicate analyses of L-glutamic acid USGS40 ($\delta^{13}C$: ±0.12‰, $\delta^{15}N$: ±0.06‰). Isotope ratios are expressed in standard $\delta$ notation (‰) relative to Pee Dee Belemnite (PDB) for carbon and atmospheric air ($N_2$) for nitrogen.

### Physiological measurements

During the 22-d experiment, photochemical efficiency was measured every 5 d on dark acclimated (20 min) fragments between 20:00 and 22:00 using an Imaging-PAM fluorometer (Waltz). Fragments were dark-acclimated for 20 min and rapid light curves were generated (RLC, 0-701 μmol m⁻² s⁻¹ PAR, 20 s intervals), using an Imaging-PAM fluorometer (MI3, SI 10, gain 2, damp 2, saturating width 0.8 s; Heinz Walz GmbH, Effeltrich, Germany). Calculations for maximal photosynthetic yield ($F_v/F_m$) and maximum relative electron transport rates ($_rETR_{max}$) were calculated according to Krueger et al.[72]. Coral skeletal (aragonite) growth measurements were taken every 7 d via the buoyant weight method[73].

After the experiment was completed, non-lyophilized subsamples of coral homogenate (90 μL) were fixed with paraformaldehyde to 4% and stored at 4 °C for symbiont density measurements. Symbiont density was determined with hemocytometer counts with a Zeiss Axioskop binocular microscope at 10–40x magnification ($n = 8$ per sample). Symbiont chlorophyll-*a*, -*c2*, and total chlorophyll concentrations were quantified spectrophotometrically after extraction in 1 ml of 90% acetone in the dark (24 h, 4 °C)[74]. The total soluble protein content of host and symbiont fractions were determined with the improved Bradford protocol, using bovine serum albumin as the protein standard[75]. Remaining airbrushed coral skeletons were then dried, cleaned of residual organic matter (10% bleach soak), and measured for surface area by wax dipping[76] to facilitate normalization of physiological parameters to surface area.

### Statistics and reproducibility

The experimental design involved collection of an average of 10 fragments from twelve parent colonies (single genotype) for 120 fragments total. Fragments were assigned numbers sequentially from collection and a random number generator was used to assign fragments to experimental treatment groups and tanks. Ten fragments were placed in each tank, with three tanks for each experimental condition. Two tanks were kept without corals for in tank POM collection (one sample per tank per timepoint). *Artemia* nauplii stock was counted three times and averaged for calculation of initial prey density. Three distinct water samples were collected from each tank after feeding to calculate post feeding prey densities.

Statistical analyses were conducted in R (version 4.2.1) and R studio (version 2022.12.0+353). The effects of feeding on physiology metrics, mass normalized and relative abundance of FAs, isotope ratios, and elemental ratios were modeled using mixed linear effects models with colony (parent genotype) and tank as orthogonal random effects to account for differences among tank conditions and colony-specific physiology. Cumulative feeding was modeled similarly but only using tank as a random effect since nauplii capture was measured at the tank level (~10 fragments tank⁻¹). We used a Tukey's Honestly Significant Difference (HSD) test for post-hoc pairwise comparisons between treatment groups within linear mixed effects models, using the 'emmeans' package in R. Feeding rate versus prey density curves were fit with a logarithmic equation with a non-zero intercept according to Ferrier-Pagès et al.[37]. Photo-physiology metrics, $F_v/F_m$ and rETR$_{max}$, were modeled as other physiology metrics but time was included as an additional fixed effect in the model. To compare experimental nutrition source groups, the biomarkers (FAs, isotopes, and elemental ratios) of the heterotrophic source (nauplii) and autotrophic source (control coral symbionts) were

compared via a Wilcoxon rank-sum test due to unequal replication between sources and deviations from normality and homogeneity for some of the biomarkers. $\Delta^{13}C$ ($\delta^{13}C_{host}$ - $\delta^{13}C_{symbiont}$) and $\Delta^{15}N$ ($\delta^{15}N_{host}$ - $\delta^{15}N_{symbiont}$) values were calculated for all fragments in which host and symbiont data were available. For isotopic data and calculations of percent element turnover due to feeding, we assumed no trophic enrichment of heavy isotopes due to little excrement of ammonia by symbiotic corals[77–79]. Percent elemental turnover for nitrogen and carbon were calculated by taking the mean isotope ratio of the host of control corals within a specific genotype (colony) and creating a mixing model between this value (0% heterotrophy) and the mean isotope value for nauplii (100% heterotrophic source). This equation is then iteratively performed for all fragments within each genotype. The equation for nitrogen and carbon turnover in host tissues with heterotrophic matter follows the format:

$$\%turnover = \left( \frac{\left( \delta_{fed\ fragment} - \delta_{control\ fragments(mean)} \right)}{\left( \delta_{nauplii(mean)} - \delta_{control\ fragments(mean)} \right)} \right) \times 100$$

## Reporting summary

Further information on research design is available in the Nature Portfolio Reporting Summary linked to this article.

## Data availability

Data is available at https://zenodo.org/records/16373686.

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

## Acknowledgements

We would like to acknowledge the ASLO LOREX program and Adina Paytan for funding travel. The Minerva Foundation (Max Planck Society) for support for M. Stuhr. The UC Santa Barbara Marine Science Analytical Laboratory for assistance in running isotope samples. We would like to thank Christie Yorke for helping run the IRMS and assistance with isotope data quality assurance checks. We would also like to thank Mia Martin, Rafael Solorzano and Iris Kern for assistance in collecting cell counts, acidifying isotope samples and running the gas chromatograph. We would like to thank Seth Newsome, Guilhem Banc-Prandi, Orit Sivan, Effrat Russel and Gilad Antler for assistance in designing the study within the Interuniversity Institute for Marine Sciences (IUI), preliminary sample checks and transportation of samples. Finally, we would like to thank all the IUI and UCSB staff that helped make this possible.

## Author contributions

C.L. and M.F. conceived of ideas. C.L., M.S., M.F., V.R., M.F. and D.V. helped design methodology. C.L. and M.S. collected data. C.L. analyzed data. C.L., M.S., M.F., V.R. and K.Mc.M. interpreted data. C.L. led the writing of the manuscript. All authors contributed critically to the drafts and gave final approval for publication.

## Competing interests

The authors declare no competing interests.
