## [Transparent Peer Review file · Communications Biology]

Selective nutrient incorporation may underestimate heterotrophy of a mixotrophic reef-building coral

Corresponding Author: Dr Connor Love

Version 0:

Reviewer comments:

Reviewer #1

(Remarks to the Author)

This is an amazing paper, and one of the best I have seen in a long time. My compliments to the authors for their detailed work! I have some suggestions below, which I hope will be addressed properly. Claims are novel and important to the field, although RQ1 is not novel, which should be addressed. The other RQ's are novel and supported by the data. Statistics seem sound, although I miss some details, such as a p value here and the lack of explaining error bars somewhere else. These are of minor concern, though. The work should be reproducible by researchers experienced with isotopic and FA analyses. Please find my comments below.

125: Do corals adjust heterotrophic feeding rates with water column food concentration? This RQ has been address several times in the literature, and the answer was always a "yes": usually, this is a type II saturating response (see work by Ferrier-Pages, and Osinga et al.). I would make it clear that RQ1 therefore is not a new question.

147 and 152: Tanks, equipped with a small pump to maximize water circulation received flow-through 130 μm filtered natural seawater. What was the flow rate of each pump? And can you estimate flow rates in cm/s around your corals? This can greatly affect feeding rates. I would also address this in the discussion.

150: Nauplii density of stock feed was counted in triplicate daily just prior to feeding. Corals were fed for 2 hrs, during which time tank flow was halted, though circulation remained to evenly distribute nauplii. How much artemia were fed initially per pulse, in artemia per liter tank water? It does not say here.

152: Triplicate water samples to assess post feeding nauplii density were taken immediately after feeding, and then tank flow was restored to flush remaining nauplii. The difference between initial and final nauplii density was used to calculate nauplii consumed per tank for each feeding, converted to biomass with an average nauplii mass of 3.2 μg (Peykaran et al., 2011). This method can be tricky, as artemia can get trapped between coral branches (or artificial structures) without actual ingestion or external digestion. That is why we chose to film our corals (Wijgerde et al. 2011). I understand that the current exp design does not allow for this.

245: what post-hoc was used?

246: I see no p value here. Mention the Bonferroni correction with $\alpha = 0.025$ as you made two post-hoc comparisons.

248: Overall, feeding rates of all fed corals varied between ~50-85 nauplii $\text{cm}^{-2} \text{hr}^{-1}$ at a prey density of ~1000 nauplii L^{-1} . Feeding rate versus prey density regressions showed that... how were you able to perform such a regression? Did you start out with different prey concentrations? It does not say in the methodology. Only that you sampled at the end.

Line 235: For isotopic data and calculations of percent element turnover due to feeding, we assumed no trophic enrichment of heavy isotopes due to little excrement of ammonia by symbiotic corals. Could the authors explain in more detail how N turnover was calculated? This could be done in the supp info to maintain paper brevity.

Figure 1: what are the error bars? SEM's, sd's?

Figure 2: please indicate significances between treatments, and explain what the graph shows. I assume box plots with medians?

Figure 3b is illegible in the word doc. I assume the online version will have a high res image? Can you explain in the caption what the legend means? Please explain the unit ranging from 0 to 1.

386: As such, the $\Delta^{13}\text{C}$ proxy "captured heterotrophy" ($\Delta^{13}\text{C} < -2\text{‰}$) in only 386 ~13% of the fragments (Fig. 4f). This could be saved for the discussion.

Figure S5: please provide a higher res version. In general, I would appreciate higher res texts in most figures.

Figure S5a: it is unclear to me how the authors arrived at the nauplii consumption shown on the x-axis. Are these totals over the exp period? I ask because the numbers are high, up to 2500 artemia per cm^2 . In table S6, the authors do refer to

cumulative artemia eaten. Perhaps this could be further clarified in figure S5.

399: replace recorded with measured or detected.

437: Bleached corals (B_F_6x) exhibited the lowest feeding rates of all treatments suggesting that there may be an energetic cost to feeding that is supplemented by the symbionts. We just completed a (yet unpublished) study showing that *Favia fragum* primary polyps consume less plankton when thermally bleached. Their tentacles become limp. It is possible that our, and the author's *Stylophora* corals, do not have sufficient ATP generation to maintain tentacle action in a bleached state due to lack of autotrophically acquired carbon, and/or that mucus secretion (also autotrophically derived) is impaired, rendering polyp tentacles less sticky and thereby reducing prey retention.

473: *Acropora* (Imbs et al., 2007; Kim, Baker, et al., 2021). Check citation.

487 to 495: please add a note on the menthol bleaching method, which is notably different from naturally occurring, thermal bleaching. These two bleaching mechanisms may act entirely different on the holobiont.

503: I don't like asking authors to cite my own work, but perhaps it would be interesting to link the author's conclusions about selective N (and P) incorporation from feeding to our findings from 2011 where we show that *Galaxea* corals may disproportionately remove N from *Artemia* nauplii. See Table 1 for potentially increased C:N and C:P ratios after external digestion (by expelled filaments) by polyps. Alas, the non-parametric test had insufficient power to prove it beyond any doubt: *J Exp Biol* (2011) 214 (20): 3351–3357. <https://doi.org/10.1242/jeb.058354>

511: ...finds that selective nutrient incorporation may be causing a considerable underestimate of the contribution of heterotrophy to corals in nature. Our study mentioned above suggests that prey clearance rate studies may also underestimate the amount of prey captured and digested. This is because (at least *Galaxea*) releases partially digested nauplii, which will be considering not captured using clearance rates. Your method will fix that! We filmed our polyps, recording what they did in detail, but you measure the end result in FA profiles. The latter method is ideal for field-collected corals. Perhaps you could briefly mention our results, and state that clearance rates in the lab, or even video analysis, may be replaced by your thorough FA profiling when estimating ex situ feeding rates.

There is no discussion about the fact that your autotrophic corals keep growing; if they could maintain this beyond the 22 days that would be quite amazing! I would devote some words to this in the discussion. The 130 µm FSW would of course still allow for feeding on bacteria and other small plankton.

Yours sincerely,

Tim Wijgerde
Marine Animal Ecology
Wageningen University and Research

Reviewer #2

(Remarks to the Author)

The manuscript titled "Selective nutrient incorporation may underestimate heterotrophy of a mixotrophic reef-building coral" presents the results of a laboratory experiment in which a gradient of heterotrophy was used to test how different markers (dN, dC and fatty acids) reflect the degree of heterotrophy in *Stylophora pistillata*. The presented experiment is very well designed, and all laboratory analyses are in line with the questions that the authors try to answer. Overall, I consider that this work is of value to the field since techniques such as stable isotope analysis are applied to natural communities with little evidence on how these values change under controlled laboratory experiments. As pointed out by the authors, the presented results expose how the use of dC values can lead to misinterpretations of the effects of environmental changes for example. The use of FA to determine heterotrophy is novel in corals and only a handful of studies have been conducted on this topic. The results presented in this manuscript suggest that FAs respond to changes in heterotrophy and hence can be utilized to study flows of nutrients in natural communities. I consider that the only thing missing in this manuscript is a quantification of heterotrophy using the different metrics. At the moment, the results only provide a visual cue of how dC, dN and FA change in the different treatments, but no mathematical approach is used to try to quantify how much heterotrophy increases with (for example) feeding. I consider that this study would greatly benefit by using mixing models to quantify host heterotrophy % and relating this value to food intake. If the authors do not agree, I would be interested in knowing why these analysis were not carried out.

Specific comments

Abstract

Lines 49-50: I consider that there is a key statement in these lines that does not get directly addressed in this study: "quantifying coral heterotrophy". Although the authors do a great job at showing how different metrics capture differences in heterotrophy across treatments, there is no direct estimation of heterotrophy. I think this is a missed opportunity, since such a controlled experiment represents the perfect opportunity for this kind of analysis (despite that this analysis is not so straightforward in nature). For example, a two-source mixing model using dN or dC values of the autotrophy and heterotrophy (as done presented in Table S3) could be used to estimate the % of heterotrophy of the host in each treatment. Then, this data could be correlated to nauplii eaten to see if there is a relationship between biomass eaten and % of heterotrophy. This approach could provide an even clearer difference between the use of dN and dC to estimate heterotrophy under natural conditions. In the case of FA this might be more complicated, but as far as I understand, Bayesian mixing models like FASTAR and mixSIAR can potentially be used to determine the % of heterotrophy using the whole FA profile of the host. This would require hands-on experience with these models, so I would recommend the authors to consult with someone with the appropriate expertise (if none of the authors is familiar with these models) to see if, given the data presented here, a quantification of heterotrophy using FA profiles is possible.

Line 51: In my opinion, by stating "We interrogated coral heterotrophy using a multi-metric biomarker approach" would mean

that all metrics were combined to determine heterotrophy. This is not the case, since each metric was used independently. Hence, I would suggest rephrasing this quote.

Introduction

Line 102: There is a typo on the sentence, please correct.

Lines 109-110: I would say that cost is the most prohibitive aspect of CSIA-AA due to the fact that the required equipment is not available at every research institution. In my experience, the EZfaast amino acid analysis kit (like the one used in your references) allows for a larger output of samples per unit time than extraction and derivatization of FAs. Hence, I would suggest removing "high time" as to a reason why to use your approach over CSIA-AA.

Materials and methods

Line 139: unit should be corrected to $\mu\text{mol quanta m}^{-2} \text{ s}^{-1}$. Same applies to Supplementary information.

Line 157: How long after the last feeding were the coral sampled for FAs?

Results

Line 246: I would suggest adding a supplementary table with the results of Figure 1A as done for the rest of the results in the manuscript. Also, was there any statistical comparison done between B_F_6x and F_6x? To the naked eye, it seems like there is not a big difference between these treatments in terms of cumulative biomass eaten (give the SD and number of replicates). So, claiming that "Coral feeding rates scale with food availability, but are significantly reduced in bleached corals" (Lines 256-257) without statistical support might not be well supported.

Line 292: The sentence is missing an 'in' between 'increases total'.

Figure 2: Why is symbiont FA presented in a different unit than symbiont protein? I would suggest presenting both metrics in $\mu\text{g/g}$ or ng/cell .

Table S3: Please disclose the unit of the data presented in the table

Line 309: I think the sentence refers to Table S4, not S3.

Line 311-312: The sentence should refer to Table S5.

Line 317: The sentence refers to Table S3, not S2.

Figure 3: I would suggest replacing Figure 3A with Figure S2. Although Fig. 3A clearly shows the difference between auto/heterotrophy, I think much more information can be acquired from Fig. S2. The differences between treatment groups gives a better understanding of how much on average they differ from each other.

Discussion

Line 429: Please correct 'the higher ...'.

Lines 437-442: Although the conclusions presented are logically possible given the results, I wonder if the bleaching with Menthol-DCMU has an effect on the corals that could explain part of the observed results. I'm not personally familiar with the method used, but I would like to hear the opinion of the authors, perhaps in your experience this bleaching protocol does not alter the physiology of corals (outside of the effect of removing the symbiont). Also, as I mentioned previously, I consider that the difference in cumulative biomass eaten between the bleached and not bleached treatment at the same feeding regime does not seem large enough (at least with no statistical testing) to claim that: "corals' ability to supplement energetic reserves with feeding and recover from bleaching may be more hindered than previously thought".

Lines 458-461: I think this sentence is too speculative. Although it is possible to degrade FAs to acetyl-CoA and used them in the Krebs cycle (TCA) to produce for example alpha-ketoglutarate and oxaloacetate, it is hard for me to believe that this happens for membrane and reserve FAs to the extent observed in the results (75% decrease). To me, it seems more likely that FAs are being used to produce energy since to perhaps compensate for the lack of an endosymbiont. Additionally, to have a clearer picture of carbon metabolism, total carbohydrate content would need to be quantified.

Lines 487-490: I think that the FA 22:6n-3 (DHA) could be mentioned here. DHA is one of the key FA biomarkers of dinoflagellates, so it is understandable that it is reduced in bleached corals. Interestingly, this FA has been linked to survival and reproduction success of many animals (same applies to 20:5n-3 or EPA), and bioconversion from shorter FAs is usually inefficient.

Version 1:

Reviewer comments:

Reviewer #1

(Remarks to the Author)

The authors have addressed my comments to my satisfaction and I recommend publishing this revised version.

Reviewer #2

(Remarks to the Author)

The authors have done a great job during the revision of the manuscript. I consider that all corrections represent an improvement over the original version, which was already of outstanding quality. The only comment I made that was not addressed was properly justified, hence I have no more comments to the work.

Referee expertise:

Referee #1: coral physiology, coral nutrition, heterotrophy

Referee #2: biochemistry, fatty acids in aquatic organisms, aquatic ecology

Reviewers' comments:

Reviewer #1 (Remarks to the Author):

This is an amazing paper, and one of the best I have seen in a long time. My compliments to the authors for their detailed work! I have some suggestions below, which I hope will be addressed properly. Claims are novel and important to the field, although RQ1 is not novel, which should be addressed. The other RQ's are novel and supported by the data. Statistics seem sound, although I miss some details, such as a p value here and the lack of explaining error bars somewhere else. These are of minor concern, though. The work should be reproducible by researchers experienced with isotopic and FA analyses. Please find my comments below.

Authors response:

125: Do corals adjust heterotrophic feeding rates with water column food concentration? This RQ has been address several times in the literature, and the answer was always a “yes”: usually, this is a type II saturating response (see work by Ferrier-Pages, and Osinga et al.). I would make it clear that RQ1 therefore is not a new question.

Authors response: We agree, we have modified the question to one of our original research questions that can be answered by our data. The revised text is copied below:

“1) Does frequency of feeding and bleached or unbleached status alter coral feeding rates?”

147 and 152: Tanks, equipped with a small pump to maximize water circulation received flow-through 130 μm filtered natural seawater. What was the flow rate of each pump? And can you estimate flow rates in cm/s around your corals? This can greatly affect feeding rates. I would also address this in the discussion.

Author's response: The flow rate of each pump was 400 L h⁻¹ from Aqua One Maxi 101 power heads and we have included this information in the aforementioned lines. We did not have any way to estimate flow rates in cm/s around the corals due to limitations of personnel and time during the experiment although we do agree this would be very helpful to understanding feeding rate changes in proportion to food supply. To the best of our abilities, we positioned the fragments evenly to homogenize small scale flow variability. The revised text is copied below:

“Tanks were equipped with a small pump (Aqua One Maxi 101, 400 L h⁻¹) to maximize water circulation and received flow-through 130 μm filtered natural seawater.”

150: Nauplii density of stock feed was counted in triplicate daily just prior to feeding. Corals were fed for 2 hrs, during which time tank flow was halted, though circulation remained to evenly distribute nauplii. How much artemia were fed initially per pulse, in artemia per liter tank water? It does not say here.

Author's response: We aimed to feed corals at a nauplii density of 1000 nauplii L⁻¹ but variations in temperature of water during egg hatching resulted in different amounts of nauplii stock to feed corals each day. To account for this, we counted nauplii stock in triplicate each day before feeding and adjusted for dilution of addition of nauplii stock into the mesocosm tanks. We have included this information in the mentioned lines and additionally we have included all the data for this as a supplemental excel file. The revised text reads as follows:

“We aimed to feed corals at a nauplii density of 1000 nauplii L⁻¹. Variations in hatching temperatures resulted in variations in total nauplii stock available, and thus nauplii stock was counted in triplicate daily just prior to feeding (see supplemental material for concentrations). Corals were fed for 2 hrs, during which time tank flow was halted, though water pumps remained on to evenly distribute nauplii. Triplicate water samples were taken immediately after feeding to assess post feeding nauplii density, and then tank flow was restored to flush remaining nauplii. The difference between initial and final nauplii density was used to calculate nauplii consumed per tank for each feeding, converted to biomass with an average nauplii mass of 3.2 μg³⁴. Due to variations in nauplii stock availability, and thus prey concentration, we were able to create prey density versus consumption rate plots and models.”

152: Triplicate water samples to assess post feeding nauplii density were taken immediately after feeding, and then tank flow was restored to flush remaining nauplii. The difference between initial and final nauplii density was used to calculate nauplii consumed per tank for each feeding, converted to biomass with an average nauplii mass of 3.2 μg (Peykaran et al., 2011). This method can be tricky, as artemia can get trapped between coral branches (or artificial structures) without actual ingestion or external digestion. That is why we chose to film our corals (Wijgerde et al. 2011). I understand that the current exp design does not allow for this.

Author's response: Yes we agree this is definitely a possibility and it would be great to include cameras in the future.

245: what post-hoc was used?

Author's response: We used a Tukey’s method for post-hoc pairwise comparisons between treatment groups using the ‘emmeans’ package in R, we have included the following sentence in the ‘statistical analysis’ portion of the methods section to reflect this detail:

“We used a Tukey’s Honestly Significant Difference (HSD) test for post-hoc pairwise comparisons between treatment groups within linear mixed effects models, using the ‘emmeans’ package in R.”

246: I see no p value here. Mention the Bonferroni correction with alpha = 0.025 as you made two post-hoc comparisons.

Author's response: Thank you for catching this, we have included the p value here from this post hoc pairwise comparison. The revised text reads as follows:

“Corals fed six times a week (F_6x) ate roughly double the nauplii (6.61 ± 1.60 mg nauplii cm^{-2}) that corals fed twice a week (F_2x) (3.11 ± 0.83 mg nauplii cm^{-2}) ($p = 0.023$) and showed larger variation in nauplii consumption among tanks (Fig 1a, Table S1). On average, bleached corals fed six times a week (B_F_6x) consumed 19% less nauplii biomass ($p = 0.500$) (5.36 ± 1.05 mg nauplii cm^{-2}) over the course of the experiment than unbleached corals fed the same amount (F_6x corals).”

248: Overall, feeding rates of all fed corals varied between ~50-85 nauplii cm^{-2} hr^{-1} at a prey density of ~1000 nauplii L^{-1} . Feeding rate versus prey density regressions showed that... how were you able to perform such a regression? Did you start out with different prey concentrations? It does not say in the methodology. Only that you sampled at the end.

Author's response: As mentioned in the above comment for line 150, we aimed to feed the corals at 1000 nauplii L^{-1} but due to variations in water temperatures during daily egg hatching we were left with variation in the density of nauplii fed to the corals each day. This allowed us to create a feeding rate vs. prey density plot because prey densities were counted before feeding and the remaining nauplii after feeding was calculated to get a feeding rate. We have included this statement in the methods for clarity:

“Due to variations in nauplii stock availability and thus prey concentration, we were able to create prey density versus consumption rate plots and models.”

Line 235: For isotopic data and calculations of percent element turnover due to feeding, we assumed no trophic enrichment of heavy isotopes due to little excrement of ammonia by symbiotic corals. Could the authors explain in more detail how N turnover was calculated? This could be done in the supp info to maintain paper brevity.

Author's response: Yes definitely, we have included the following details in the supplement for clarity:

“Percent elemental turnover for nitrogen and carbon were calculated by taking the mean isotope ratio of the host of control corals within a specific genotype (colony) and creating a mixing model between this value (0% heterotrophy) and the mean isotope value for nauplii (100% heterotrophic source). This equation is then iteratively performed for all fragments within each genotype. The equation for nitrogen and carbon turnover in host tissues with heterotrophic matter follows the format:

$$\% \text{ turnover} = \frac{(\text{fed fragment} - \text{control fragments (mean)})}{(\text{nauplii (mean)} - \text{control fragments (mean)})} * 100$$

Figure 1: what are the error bars? SEM's, sd's?

Author's response: Thank you for pointing this out, the error bars are standard error. We have included this detail in the figure caption.

Figure 2: please indicate significances between treatments, and explain what the graph shows. I assume box plots with medians?

Author's response: We have placed pair-wise comparison stats in the figure panels and inserted additional information for the boxplot. The text now reads:

*“Physiology metrics by experimental treatment (bleached + fed 6x week⁻¹: pink, control: green, fed 2x week⁻¹: yellow, fed 6x week⁻¹: blue) as median box and whisker plots with raw data added and connecting bars indicating significance levels of post hoc pairwise contrasts: *p < 0.05, **p < 0.01, ***p < 0.001.”*

Figure 3b is illegible in the word doc. I assume the online version will have a high res image? Can you explain in the caption what the legend means? Please explain the unit ranging from 0 to 1.

Author's response: We believe this may be due to the image being compressed when the manuscript was transferred, we will be submitting full resolution images with our edits. Additionally, we have put in an explanation for the legend of 3b as well, it now reads:

“b) a heatmap of min-max normalized FA data (blue to red scale bar from 0 [min] to 1 [max]) for all corals, grouped by experimental source group on the y-axis and observed pattern of individual FAs on the x-axis.”

386: As such, the $\Delta^{13}\text{C}$ proxy “captured heterotrophy” ($\Delta^{13}\text{C} < -2\text{‰}$) in only 386 ~13% of the fragments (Fig. 4f). This could be saved for the discussion.

Author's response: Done.

Figure S5: please provide a higher res version. In general, I would appreciate higher res texts in most figures.

Author's response: We are unsure why this is low res in your view but will ensure that higher resolution images get transferred to you in the second round of edits.

Figure S5a: it is unclear to me how the authors arrived at the nauplii consumption shown on the x-axis. Are these totals over the exp period? I ask because the numbers are high, up to 2500 artemia per cm². In table S6, the authors do refer to cumulative artemia eaten. Perhaps this could be further clarified in figure S5.

Author's response: Yes this refers to cumulative nauplii consumption over the course of the whole experiment. We have included this sentence in the figure caption for clarity:

“a) Nitrogen turnover versus cumulative nauplii consumed over the entire experiment.”

399: replace recorded with measured or detected.

Author's response: We have replaced this word with “measured” as suggested.

437: Bleached corals (B_F_6x) exhibited the lowest feeding rates of all treatments suggesting that there may be an energetic cost to feeding that is supplemented by the symbionts. We just completed a (yet unpublished) study showing that *Favia fragum* primary polyps consume less plankton when thermally bleached. Their tentacles become limp. It is possible that our, and the author's *Stylophora* corals, do not have sufficient ATP generation to maintain tentacle action in a bleached state due to lack of autotrophically acquired carbon, and/or that mucus secretion (also autotrophically derived) is impaired, rendering polyp tentacles less sticky and thereby reducing prey retention.

Author's response: That is very interesting and we agree that it is possible these bleached corals were lacking ATP since stored carbohydrates are used to generate ATP in these corals (Kochman et al., 2021) and the symbionts transfer a great deal of carbohydrates to the host (Falkowski et al., 1984). We have included a brief statement to reflect this possibility:

*“Bleached corals (B_F_6x) exhibited the lowest feeding rates of any treatment (Fig. 1b), even though cumulative biomass consumption was not statistically different (Fig. 1a, see Results). That said, mean cumulative biomass consumption of bleached corals was still lower than non-bleached corals fed the same amount (Fig. 1a). Altogether, this suggests (with soft evidence) that some corals can exhibit reduced feeding rates after bleaching⁵⁵ and that there may be an energetic cost to feeding that is supplemented by the symbionts. One potential explanation is that since Red Sea *S. pistillata* can generate ATP from stored carbohydrates⁵⁶ that are mainly generated by the symbiont⁷, this would reduce the energetic pool available to bleached corals for feeding activities (e.g., tentacle movements).”*

473: *Acropora* (Imbs et al., 2007; Kim, Baker, et al., 2021). Check citation.

Author's response: Noted, we have updated the citations.

487 to 495: please add a note on the menthol bleaching method, which is notably different from naturally occurring, thermal bleaching. These two bleaching mechanisms may act entirely different on the holobiont.

Author's response: We have added a note on menthol induced bleaching before the lines mentioned here (in the physiology section of the discussion) so as to define our interpretations from there and onward. The note reads:

*“Although it is known that the menthol induced bleaching approach we used does result in different mechanisms of symbiont removal than thermal bleaching in anenomes⁵⁸, the physiological and biochemical performance of *S. pistillata* does not significantly change with menthol bleaching compared to unbleached corals³³. As such, we interpret patterns of our bleached and fed corals (B_F_6x) to stem primarily from the observed low symbiont densities*

(Fig. 2g, Table S2), low photo-physiological performance (Fig. 2f, Table S3), and cumulative nauplii biomass consumed (Fig. 1a, Table S1) rather than bleaching method.”

503: I don't like asking authors to cite my own work, but perhaps it would be interesting to link the author's conclusions about selective N (and P) incorporation from feeding to our findings from 2011 where we show that Galaxea corals may disproportionately remove N from Artemia nauplii. See Table 1 for potentially increased C:N and C:P ratios after external digestion (by expelled filaments) by polyps. Alas, the non-parametric test had insufficient power to prove it beyond any doubt: J Exp Biol (2011) 214 (20): 3351–3357. <https://doi.org/10.1242/jeb.058354>

Author's response: This is a great finding that definitely fits into the findings of our manuscript, we have cited this 2011 paper here.

511: ...finds that selective nutrient incorporation may be causing a considerable underestimate of the contribution of heterotrophy to corals in nature. Our study mentioned above suggests that prey clearance rate studies may also underestimate the amount of prey captured and digested. This is because (at least Galaxea) releases partially digested nauplii, which will be considering not captured using clearance rates. Your method will fix that! We filmed our polyps, recording what they did in detail, but you measure the end result in FA profiles. The latter method is ideal for field-collected corals. Perhaps you could briefly mention our results, and state that clearance rates in the lab, or even video analysis, may be replaced by your thorough FA profiling when estimating ex situ feeding rates.

Author's response: We agree this is a very interesting finding and definitely does relate to these results, however our work is primarily about the decoupling of biomarker tracers that are most commonly used for food web studies and the ramifications this has when we interpret biomarker patterns from nature. This particular line you mention is in the last paragraph where we are summarizing all of our observed patterns and we have cited your work here which is a critical point in the paper where we are mentioning selective nitrogen incorporation from heterotrophy and trust that readers will follow this citation to your paper.

There is no discussion about the fact that your autotrophic corals keep growing; if they could maintain this beyond the 22 days that would be quite amazing! I would devote some words to this in the discussion. The 130 μ m FSW would of course still allow for feeding on bacteria and other small plankton.

Author's response: Agreed, we have included a mention of this in the physiological section of the discussion. The text now reads:

“Skeletal growth also positively scaled with heterotrophy; and was net even positive for unfed control corals (Fig. 2e), perhaps due to the resilient nature of Red Sea S. pistillata⁵⁷, short experiment duration (3 weeks), and/or some amount of heterotrophy on $\leq 130\text{-}\mu\text{m}$ particles let in by the sweater supply system (see Methods).”

Yours sincerely,

Tim Wijgerde
Marine Animal Ecology
Wageningen University and Research

Reviewer #2 (Remarks to the Author):

The manuscript titled “Selective nutrient incorporation may underestimate heterotrophy of a mixotrophic reef-building coral” presents the results of a laboratory experiment in which a gradient of heterotrophy was used to test how different markers (dN, dC and fatty acids) reflect the degree of heterotrophy in *Stylophora pistillata*. The presented experiment is very well designed, and all laboratory analyses are in line with the questions that the authors try to answer. Overall, I consider that this work is of value to the field since techniques such as stable isotope analysis are applied to natural communities with little evidence on how these values change under controlled laboratory experiments. As pointed out by the authors, the presented results expose how the use of dC values can lead to misinterpretations of the effects of environmental changes for example. The use of FA to determine heterotrophy is novel in corals and only a handful of studies have been conducted on this topic. The results presented in this manuscript suggest that FAs respond to changes in heterotrophy and hence can be utilized to study flows of nutrients in natural communities. I consider that the only thing missing in this manuscript is a quantification of heterotrophy using the different metrics. At the moment, the results only provide a visual cue of how dC, dN and FA change in the different treatments, but no mathematical approach is used to try to quantify how much heterotrophy increases with (for example) feeding. I consider that this study would greatly benefit by using mixing models to quantify host heterotrophy % and relating this value to food intake. If the authors do not agree, I would be interested in knowing why these analysis were not carried out.

Authors response: Thank you for taking the time to read our manuscript and suggest edits. As it stands, we think it would be misleading and potentially erroneous to conduct a % heterotrophy calculation from these FA data since we have no calibration coefficients from a 100% turnover experiment prior to this. To quantify dietary inputs for fatty acids, one needs known calibration coefficients for all or at least several fatty acids (Guerrero and Rogers, 2020) which are unique for each distinct permutation of consumer and diet source and to our knowledge this does not exist for our specific consumer/diet system. Alternatively, one could transform the fatty acid relative abundances of the diet source into a consumer fed entirely that source to ~100% FA turnover and then plug it into MixSIAR (again see Guerrero and Rogers or Galloway et al., 2015). To do this, one needs to conduct a feeding experiment of the organism of interest in which there is a ~100% turnover of consumer tissue with a diet source. Given time constraints of our study (~3 week experiment) and known turnover rates of corals (at least 2 months for the fastest turnover rates), alongside our data showing ~45% nitrogen turnover of host tissue with nauplii biomass in the high feed treatments, we feel we would be knowingly calculating incorrect answers.

Instead, we suggest that these data, and data from other papers would better be explored in a comparative paper in which many methods of calculating % heterotrophy for mixotrophs are

explored and compared across several studies. We think this topic is extremely interesting and important but is also an entire paper unto itself. We feel that trying to tackle that comparative analysis would take away from the main story of this manuscript, which is that there appears to be a coupled intake of certain fatty acids and nitrogen but variable and decoupled intake of elemental carbon.

Although we did not calculate a global percent heterotrophy, we did calculate % turnover of host tissue with heterotrophic carbon and nitrogen as separate elements. We did this from mixing models in which the host of unfed corals acts as one end member and the nauplii act as the other endmember (assuming no trophic discrimination factor which is common for corals). We then plotted host heterotrophic nitrogen turnover with cumulative prey capture (Figure S4) and compared the two in the results and discussion. Additionally, we fit nitrogen turnover to michaelis-menten hyperbolic equations for three fatty acids to provide mathematical evidence for the coupled intake of nitrogen and select fatty acids. We think this is a very precise way to look at heterotrophy and supports one of the main findings of our manuscript that carbon and nitrogen incorporation from heterotrophy can be severely decoupled, which calls into question the interpretations of some carbon isotope proxies or coupled carbon and nitrogen data when corals are eating at lower trophic levels (phytoplankton in water column) such that there is not a sufficient difference in nitrogen isotopes between nutritional source groups to detect the shift.

Specific comments

Abstract

Lines 49-50: I consider that there is a key statement in these lines that does not get directly addressed in this study: “quantifying coral heterotrophy”. Although the authors do a great job at showing how different metrics capture differences in heterotrophy across treatments, there is no direct estimation of heterotrophy. I think this is a missed opportunity, since such a controlled experiment represents the perfect opportunity for this kind of analysis (despite that this analysis is not so straightforward in nature). For example, a two-source mixing model using δN or δC values of the autotrophy and heterotrophy (as done presented in Table S3) could be used to estimate the % of heterotrophy of the host in each treatment. Then, this data could be correlated to nauplii eaten to see if there is a relationship between biomass eaten and % of heterotrophy. This approach could provide an even clearer difference between the use of δN and δC to estimate heterotrophy under natural conditions. In the case of FA this might be more complicated, but as far as I understand, Bayesian mixing models like FASTAR and mixSIAR can potentially be used to determine the % of heterotrophy using the whole FA profile of the host. This would require hands-on experience with these models, so I would recommend the authors to consult with someone with the appropriate expertise (if none of the authors is familiar with these models) to see if, given the data presented here, a quantification of heterotrophy using FA profiles is possible.

Author’s response: Thank you for your comment but we do not directly say that we will address “quantifying coral heterotrophy” within this sentence. The sentence reads: “*Heterotrophy promotes recovery and resistance to environmental stress, but quantifying coral heterotrophy remains difficult due to complex resource exchanges within the coral holobiont.*” We are merely emphasizing the fact that while many studies attempt to assess heterotrophy, the underlying factors controlling the variables used to assess heterotrophy are not well constrained, making accurate quantification difficult.

To get around the complexities (and potentially misleading results) of heterotrophy quantification, we instead calculate % turnover of host nitrogen (and carbon) with heterotrophic nitrogen (and carbon) using a mixing model. Further, we plot % turnover of nitrogen with heterotrophic nitrogen with nauplii eaten and different fatty acids and mathematically fit these data to hyperbolic Michaelis-Menten equations in Figure S4. We found that carbon isotopes, nitrogen isotopes, and fatty acids all tell a different story about coral metabolism. Nitrogen and select fatty acids appear to be coupled while carbon is highly variable and decoupled in this system. In this way “% heterotrophy” of corals has many different stories depending on how it is quantified. By focusing more on the chemical currency of elemental incorporation we believe that our study presents a more transparent estimate of heterotrophic nutrition. See response to question above for information on why trying to calculate % heterotrophy on FA data is likely inappropriate for this study.

Line 51: In my opinion, by stating “We interrogated coral heterotrophy using a multi-metric biomarker approach” would mean that all metrics were combined to determine heterotrophy. This is not the case, since each metric was used independently. Hence, I would suggest rephrasing this quote.

Authors response: Agreed, we have changed the sentence to read:

“We interrogated the response of multiple biomarkers to coral heterotrophy using fatty acid profiling and stable isotope analysis of Stylophora pistillata grown along a controlled feeding gradient from pure autotrophy to pure heterotrophy.”

Introduction

Line 102: There is a typo on the sentence, please correct.

Author’s response: We are unable to find the typo that the reviewer alluded to. Without more details, we are unable to make any changes in this regard.

Lines 109-110: I would say that cost is the most prohibitive aspect of CSIA-AA due to the fact that the required equipment is not available at every research institution. In my experience, the EZfaast amino acid analysis kit (like the one used in your references) allows for a larger output of samples per unit time than extraction and derivatization of FAs. Hence, I would suggest removing “high time” as to a reason why to use your approach over CSIA-AA.

Author’s response: Agreed. We have changed the sentence to read:

“While some studies have used compound-specific amino acid isotope analysis (CSIA-AA) to clarify coral trophic strategies^{12,23}, the cost and limited availability of instrumentation for such analyses can be prohibitive and lead to low sample throughput.”

Materials and methods

Line 139: unit should be corrected to $\mu\text{mol quanta m}^{-2} \text{ s}^{-1}$. Same applies to Supplementary information.

Author's response: Thank you for this catch we have changed this in the methods in the main manuscript and in the Supplement.

Line 157: How long after the last feeding were the coral sampled for FAs?

Author's response: Corals were sampled 24 hours after the last feeding. We have included a statement at line 157 to reflect this. The modified texts reads:

“Corals were flash frozen in liquid nitrogen ~24 hrs after the last feeding and airbrushed with 10 mL of cold phosphate buffer (0.1 M) with EDTA (0.1 mM, pH = 7.0) at 4°C before being manually homogenized on ice.”

Results

Line 246: I would suggest adding a supplementary table with the results of Figure 1A as done for the rest of the results in the manuscript. Also, was there any statistical comparison done between B_F_6x and F_6x? To the naked eye, it seems like there is not a big difference between these treatments in terms of cumulative biomass eaten (give the SD and number of replicates). So, claiming that “Coral feeding rates scale with food availability, but are significantly reduced in bleached corals” (Lines 256-257) without statistical support might not be well supported.

Author's response: We have provided a supplementary table for the results of Figure 1a as Table S1 and have changed subsequent supplementary table numbers with the new order. Additionally, we have plotted raw data with the mean (+ or - SD) in Figure 1a to be in line with nature's requests for figure formatting. A post hoc statistical comparison was done between B_F_6x and F_6x and we have provided this in the first section of the results, the result is non-significant ($p = 0.5$). However, we do have significant slope estimates for logarithmic feeding rate models (see Results and Figure 2b). So while cumulative biomass capture is not statistically significant, when we consider all feeding data from every day during the course of the experiment (Figure 2b) the model slope estimates are highly significant and different between groups (B_F_6x: $a = 33.8$, $p < 0.001$; F_6x: $b = 41$, $p < 0.001$). Due to this contrast in evidence we have heavily softened our claims within the discussion, called for additional research on the topic and have changed the Figure caption title so that it no longer claims significance, it now reads: “Coral feeding rates scale with food availability, but are reduced in bleached corals.” The modified discussion text now reads:

*“Bleached corals (B_F_6x) exhibited the lowest feeding rates of any treatment (Fig. 1b), even though cumulative biomass consumption was not statistically different (Fig. 1a, see Results). That said, mean cumulative biomass consumption of bleached corals was still lower than non-bleached corals fed the same amount (Fig. 1a). Altogether, this suggests (with soft evidence) that some corals can exhibit reduced feeding rates after bleaching⁵⁵ and that there may be an energetic cost to feeding that is supplemented by the symbionts. One potential explanation is that since Red Sea *S. pistillata* can generate ATP from stored carbohydrates⁵⁶ that are mainly generated by the symbiont⁷, this would reduce the energetic pool available to bleached corals for feeding activities (e.g., tentacle movements). Although our results are contrasting, reduced feeding rates of bleached corals would have further implications in the face of a warming global*

ocean. Here, bleached corals would consume less plankton than unbleached corals given the same heterotrophic food supplies, further amplifying the negative effects of coral bleaching on some coral species. This is a hypothesis ripe for further investigation.”

Line 292: The sentence is missing an ‘in’ between ‘increases total’.

Author's response: Thank you for this catch, we have put an ‘in’ here.

Figure 2: Why is symbiont FA presented in a different unit than symbiont protein? I would suggest presenting both metrics in ug/g or ng/cell.

Author's response: Agreed, we have put both units in ug/g. Please see edited Figure 2.

Table S3: Please disclose the unit of the data presented in the table

Author's response: Thank you for this catch, we have included units in the table caption as follows:

“Table shows mean and standard deviation fatty acid data (% total), isotope ratios (permil) and one elemental ratio (no units) of both autotrophic (control coral symbionts) and heterotrophic (nauplii) nutritional source groups during the experiment.”

Line 309: I think the sentence refers to Table S4, not S3.

Author's response: Thank you, this has been changed.

Line 311-312: The sentence should refer to Table S5.

Author's response: Thank you, a reference to Table S5 has been added here.

Line 317: The sentence refers to Table S3, not S2.

Author's response: This has been changed to Table S3.

Figure 3: I would suggest replacing Figure 3A with Figure S2. Although Fig. 3A clearly shows the difference between auto/heterotrophy, I think much more information can be acquired from Fig. S2. The differences between treatment groups gives a better understanding of how much on average they differ from each other.

Author's response: We agree, we have changed Figure 3a with Figure S2 and think it provides more information on this figure and the story as a whole.

Discussion

Line 429: Please correct ‘the higher ...’.

Author's response: Done.

Lines 437-442: Although the conclusions presented are logically possible given the results, I wonder if the bleaching with Menthol-DCMU has an effect on the corals that could explain part of the observed results. I'm not personally familiar with the method used, but I would like to hear the opinion of the authors, perhaps in your experience this bleaching protocol does not alter the physiology of corals (outside of the effect of removing the symbiont). Also, as I mentioned previously, I consider that the difference in cumulative biomass eaten between the bleached and not bleached treatment at the same feeding regime does not seem large enough (at least with no statistical testing) to claim that: "corals' ability to supplement energetic reserves with feeding and recover from bleaching may be more hindered than previously thought".

Author's response: To answer the first concern, yes we have definitely considered adverse effects of menthol bleaching as a possibility but previous work has shown that the physiological and biochemical performance of *Stylophora pistillata* remains unchanged after menthol induced bleaching (see Wang et al., 2012;

<https://journals.plos.org/plosone/article?id=10.1371/journal.pone.0046406>).

To answer the second concern, we agree that the cumulative biomass consumed is not large enough to claim a difference between bleached and unbleached corals (B_F_6x and F_6x treatments) and we do conduct statistical tests in the first section of the results, however, we did see significantly reduced feeding *rates* calculated from non-linear regressions slope estimates (see Results and Figure 1b). We have revised this statement to more accurately reflect our results:

*"Bleached corals (B_F_6x) exhibited the lowest feeding rates of any treatment (Fig. 1b), even though cumulative biomass consumption was not statistically different (Fig. 1a, see Results). That said, mean cumulative biomass consumption of bleached corals was still lower than non-bleached corals fed the same amount (Fig. 1a). Altogether, this suggests (with soft evidence) that some corals can exhibit reduced feeding rates after bleaching⁵⁵ and that there may be an energetic cost to feeding that is supplemented by the symbionts. One potential explanation is that since Red Sea *S. pistillata* can generate ATP from stored carbohydrates⁵⁶ that are mainly generated by the symbiont⁷, this would reduce the energetic pool available to bleached corals for feeding activities (e.g., tentacle movements). Although our results are contrasting, reduced feeding rates of bleached corals would have further implications in the face of a warming global ocean. Here, bleached corals would consume less plankton than unbleached corals given the same heterotrophic food supplies, further amplifying the negative effects of coral bleaching on some coral species. This is a hypothesis ripe for further investigation."*

Lines 458-461: I think this sentence is too speculative. Although it is possible to degrade FAs to acetyl-CoA and used them in the Krebs cycle (TCA) to produce for example alpha-ketoglutarate and oxaloacetate, it is hard for me to believe that this happens for membrane and reserve FAs to the extent observed in the results (75% decrease). To me, it seems more likely that FAs are being used to produce energy since to perhaps compensate for the lack of an endosymbiont. Additionally, to have a clearer picture of carbon metabolism, total carbohydrate content would need to be quantified.

Author's response: We agree, we have added this explanation to the lines mentioned above as the most plausible and additionally have found evidence that Red Sea *Stylophora pistillata* collected from the same site does indeed catabolize lipids during bleaching stress, we will put this as the primary explanation but will also leave routing of some fatty acids into amino acids as a plausible explanation/ contributor which was seen in Treignier et al., 2008. The new sentence reads as follows:

“Interestingly, we found that total FA mass of B_F_6x host tissues decreased more than total protein (75% vs 26%, respectively). This likely indicates that bleached corals were preferentially catabolizing FAs for energy to compensate for low symbiont densities^{17,67,68}, but may also include some routing of lipid reserves into amino acid synthesis⁶⁵.”

Lines 487-490: I think that the FA 22:6n-3 (DHA) could be mentioned here. DHA is one of the key FA biomarkers of dinoflagellates, so it is understandable that it is reduced in bleached corals. Interestingly, this FA has been linked to survival and reproduction success of many animals (same applies to 20:5n-3 or EPA), and bioconversion from shorter FAs is usually inefficient.

Author's response: Yes definitely, 22:6n3 was also one of our “bleaching pattern” FAs in Figure 3. We will include mention of this here. Interestingly, EPA follows patterns as a “heterotrophic” FA from our dataset but might be primarily due to the fact that the nauplii were higher in this FA than the symbionts (~5.5% vs. 2.5% of total) and the corals might just incorporate it from wherever it is in higher proportions. We have modified the sentence and it now reads:

“14:0, 16:1n7, 18:4n3, 20:4n3 and 22:6n3 showed dramatic declines in the B_F_6x treatment (Table S4), which is in line with evidence that the PUFAs in this list are mainly sourced from the symbionts⁷⁶. The sourcing of 14:0 and 16:1n7 from the symbionts is less defined. Nonetheless, these FAs are often found in higher proportions in the symbionts than the host of several species⁶¹ (Table S5 and S6) and follow patterns of symbiont sourcing (Fig. 3b).”